# Multiclass Classification for Hawkes Processes

**Christophe Denis**[1]             **Charlotte Dion-Blanc**[2]             **Laure Sansonnet**[3]

[1]LAMA, Université Gustave Eiffel, France
[2]LPSM, Sorbonne Université, France
[3]AgroParisTech, MIA-Paris-Saclay, Université Paris-Saclay, France

## Abstract

We investigate the multiclass classification problem where the features are event sequences. More precisely, the data are assumed to be generated by a mixture of simple linear Hawkes processes. In this new setting, the classes are discriminated by various triggering kernels. A challenge is then to build an efficient classification procedure. We derive the optimal Bayes rule and provide a two-step estimation procedure of the Bayes classifier. In the first step, the weights of the mixture are estimated; in the second step, an empirical risk minimization procedure is performed to estimate the parameters of the Hawkes processes. We establish the consistency of the resulting procedure and derive rates of convergence. Finally, the numerical properties of the data-driven algorithm are illustrated through a simulation study where the triggering kernels are assumed to belong to the popular parametric exponential family. It highlights the accuracy and the robustness of the proposed algorithm. In particular, even if the underlying kernels are misspecified, the procedure exhibits good performance.

## 1 INTRODUCTION

A crucial challenge in multiclass learning is to provide algorithms designed to handle temporal data. In the present paper, we tackle the multiclass classification problem where the features are time event sequences. More precisely, we assume that the data come from a mixture of Hawkes processes and we focus on the classification per trajectory (and not per event).

In neuroscience, we can consider event sequences as recorded spike trains on several neurons from different populations (healthy or sick subjects, for instance). The goal is then to predict the status (healthy or not) of a new subject from the associated recording [Lambert et al., 2018].

Hawkes processes, originally introduced in [Hawkes, 1971], are proposed to model tricky event sequences where the past events influence the future events. Hawkes processes arise in a wide variety of fields, ranging from neuroscience to finance. In mathematical finance, see *e.g.* [Bacry et al., 2015] for a complete review; in the social network literature, see *e.g.* [Lukasik et al., 2016] and [Qu and Lemhadri, 2021]. In neuroscience, Hawkes processes have a statistical interest for modeling neuron spike occurrences, see *e.g.* [Hansen et al., 2015], [Ditlevsen and Löcherbach, 2017], [Foschi, 2020].

Seminal work for Hawkes process properties is [Brémaud and Massoulié, 1996]. Furthermore, there are numerous statistical methods of inference for Hawkes processes. For instance, one can cite [Hansen et al., 2015], [Bacry and Muzy, 2016] and more recently [Bacry et al., 2020], or in a Bayesian framework, [Rasmussen, 2013]. Besides, [Favetto, 2019] focuses on parameter estimation for Hawkes processes from repeated observations in the context of electricity market modeling.

However, the aim of the paper is a multiclass classification task and not the parameter inference. To the best of our knowledge, except the paper of [Lukasik et al., 2016], there is no work which deals with supervised classification for Hawkes processes. In [Lukasik et al., 2016], the authors propose to use multivariate Hawkes processes for classifying sequences of temporal textual data, with an application to rumours coming from Twitter datasets. They highlight that a model based on Hawkes processes is a competitive approach which takes into account the temporal dynamic of the data. But, they do not provide any theoretical properties.

More recently, Dutta et al. [2020], Tondulkar et al. [2020], Ram and Srijith [2018] focus on the question of time classification. Indeed, for classical Twitter example from PHEME dataset, used to do rumor stance classification, the models impose a label on each tweet (each time). The classification setting, with temporal and textual data, is thus a bit different

*Accepted for the 38th Conference on Uncertainty in Artificial Intelligence* (UAI 2022).

from our framework. Besides, as in [Lukasik et al., 2016], the authors do not provide theoretical properties to support their procedures.

In this work, we observe repeatedly jump times coming from the mixture of Hawkes processes, on a fixed time interval $[0, T]$. The classes are characterized by different triggering kernels. We first formally define the model and provide the explicit form of the Bayes classifier in Section 2. The expression of the Bayes classifier suggests to consider a plug-in approach to estimate the optimal predictor. Section 3 is devoted to the definition of plug-in type classifier and the study of its properties. We show how the misclassification error, for any plug-in predictor is linked to the estimation error of the process parameters. We propose in Section 4 a two-step procedure to build a plug-in type classifier. A first step is dedicated to the estimation of the weights of the mixture. In a second step the parameters of the process are estimated through an empirical risk minimization procedure by using similar ideas as in [Denis et al., 2020]. The resulting algorithm benefits from the attractive properties of the empirical risk minimizer: it is computationally efficient and offers good theoretical properties. In particular, under mild assumptions, we show that the proposed procedure performs as well as the Bayes classifier. Section 5 illustrates the performance and the robustness of the method in the case where the triggering kernels are assumed to belong to the parametric exponential family. Finally, a discussion which highlights some directions for future works is proposed in Section 6.

## 2 GENERAL FRAMEWORK

Section 2.1 introduces the considered model, some notation and explains the objective of the paper. In Section 2.2, we provide an explicit formula of the optimal predictor.

### 2.1 STATISTICAL SETTING

Let $Y$ a random variable which takes its values in $\mathcal{Y} = \{1, \ldots, K\}$, with $K \geq 2$, representing the label of the observations. The distribution of $Y$ is denoted by $\mathbf{p}^* = (p_k^*)_{k \in \mathcal{Y}}$ and is unknown. We assume that the observations come from a mixture $N$ of simple linear Hawkes processes observed on the time interval $[0, T]$. Precisely, conditionally on $Y$, $N$ is a simple linear Hawkes process. The number of points that lie in $[0, t]$ is denoted by $N_t$ and the corresponding counting process is $(N_t)_{0 \leq t \leq T}$. The jump times of $N$ are denoted $T_1, \ldots, T_{N_T}$. The filtration (or history) at time $t^-$ is denoted $\mathcal{F}_{t^-}$ and contains all the necessary information for generating the next point of $N$.

**Conditional intensity** The intensity of the process $N$ at time $t \geq 0$, with respect to the filtration $(\mathcal{F}_t)_{t \geq 0}$, is defined

as

$$\lambda_Y^*(t) := \lambda^{(\mu^*, h_Y^*)}(t) := \mu^* + \sum_{T_i < t} h_Y^*(t - T_i), \quad (1)$$

where the first term $\mu^* > 0$ is the baseline, or exogenous intensity, and the second term is a weighted sum over past events. For each class $k \in \mathcal{Y}$, the function $h_k^*$ is the triggering kernel which is nonnegative and supported on $\mathbb{R}_+$. Besides, both parameters $\mu^*$ and $\mathbf{h}^* = (h_1^*, \ldots, h_K^*)$ are assumed to be unknown.

Note that the baseline intensity is assumed to be common to all classes. This assumption is notwithstanding consistent according to the neuronal experimental setting described in Section 1. Indeed, if the spike trains are recorded on the same type of neurons (*e.g.* neurons which play the same role), it seems relevant to assume that the exogenous intensity is homogeneous between the classes.

**Objective** Given a sequence $\mathcal{T}_T = \{T_1, \ldots, T_{N_T}\}$ of observed jump times of $N$ over the fixed interval $[0, T]$, the goal is then to build a predictor, namely a classifier $g$, a measurable function such that $g(\mathcal{T}_T)$ is a prediction of the associated label $Y$. The performance of a classifier $g$ is then measured through its misclassification risk

$$\mathcal{R}(g) := \mathbb{P}(g(\mathcal{T}_T) \neq Y).$$

In the following, we denote by $\mathcal{G}$ the set of classifiers.

### 2.2 BAYES RULE

The unknown minimizer of $\mathcal{R}$ over $\mathcal{G}$ is the so-called Bayes classifier, denoted by $g^*$, and is characterized by

$$g^*(\mathcal{T}_T) \in \underset{k \in \mathcal{Y}}{\operatorname{argmax}} \, \pi_k^*(\mathcal{T}_T),$$

with $\pi_k^*(\mathcal{T}_T) = \mathbb{P}(Y = k | \mathcal{T}_T)$. The following proposition gives the expression of the conditional probabilities $\pi_k^*$ and then provides a closed form of the Bayes classifier.

**Proposition 2.1.** *Let $T \geq 0$. For each $k \in \mathcal{Y}$, we define,*

$$F_k^*(\mathcal{T}_T) = F^{(\mu^*, h_k^*)}(\mathcal{T}_T) \quad (2)$$
$$:= -\int_0^T \lambda^{(\mu^*, h_k^*)}(s) \, \mathrm{d}s + \sum_{T_i \in \mathcal{T}_T} \log(\lambda^{(\mu^*, h_k^*)}(T_i)).$$

*Therefore, the sequence of conditional probabilities satisfies*

$$\pi_k^*(\mathcal{T}_T) = \phi_k^{\mathbf{p}^*}(\mathbf{F}^*(\mathcal{T}_T)) \quad \mathbb{P} - a.s.,$$

*where $\mathbf{F}^* = (F_1^*, \ldots, F_K^*)$ and $\phi_k^{\mathbf{p}^*} : (x_1, \ldots, x_K) \mapsto$*
$$\frac{p_k^* \mathrm{e}^{x_k}}{\sum_{j=1}^K p_j^* \mathrm{e}^{x_j}} \text{ are softmax functions.}$$

Note that conditionally on the event $Y = k$, $F_k^*(\mathcal{T}_T)$ is the likelihood function of the sequence $\mathcal{T}_T$. Proposition 2.1 highlights the dependencies of the optimal Bayes classifier *w.r.t.* the unknown parameters. In the following, for a given classifier $g \in \mathcal{G}$, we define its excess risk as

$$\mathcal{E}(g) := \mathcal{R}(g) - \mathcal{R}(g^*).$$

# 3 PLUG-IN TYPE CLASSIFIER

We first introduce assumptions related to the model in Section 3.1 and then define a set of classifiers which relies on the plug-in principle in Section 3.2. Finally, the main properties of the plug-in classifier are provided in Section 3.3.

## 3.1 ASSUMPTIONS

We first make the following assumptions on the triggering kernels.

**Assumption 3.1** (Stability condition)**.** *For each $k \in \mathcal{Y}$, $h_k : \mathbb{R}_+ \to \mathbb{R}_+$ is bounded and satisfies $\int h_k(t) \, \mathrm{d}t < 1$.*

**Assumption 3.2.** *There exist $0 < \mu_0 < \mu_1$ such that $\mu_0 \leq \mu^* \leq \mu_1$.*

**Assumption 3.3.** *There exists a positive constant $p_0$ such that $\min(\mathbf{p}^*) > p_0$.*

Assumption 3.1 guarantees that $N_T$ admits finite exponential moments, that is, there exists $a > 0$ such that $\mathbb{E}[\exp(a|N_T|)] < \infty$, see for instance [Roueff et al., 2016]. In particular the exponential and power-law kernels satisfy this assumption (with additional assumptions on the corresponding parameters). Assumption 3.2 is a technical assumption and Assumption 3.3 ensures that all the components of the mixture occur with non-zero probability.

Let us denote the following subset of probability weights

$$\mathcal{P}_{p_0} := \{\mathbf{p} \in \mathbb{R}_+^K : \sum_{i=1}^K p_i = 1, \ \min(\mathbf{p}) > p_0\}.$$

## 3.2 DEFINITIONS

In this section, we present the construction of the plug-in type classifiers.

First we introduce a set $\mathcal{H}$ of nonnegative functions supported on $\mathbb{R}_+$. For a $K$-tuple $\mathbf{h} = (h_1, \dots, h_K)$ in $\mathcal{H}^K$, we associate $\mathbf{p}$ a vector of probability weights and a baseline intensity $\mu > 0$. For each $k \in \mathcal{Y}$, we then define

$$\lambda_k(t) = \lambda^{(\mu, h_k)}(t) = \mu + \sum_{T_i < t} h_k(t - T_i), \quad t \in [0, T].$$

Hence, the random functions $(\lambda_k)_{k=1,\dots,K}$ are approximations of the conditional intensities $\lambda_k^*$ defined by (1). Besides, similarly with the definition (2) of $F_k^*(\mathcal{T}_T)$, we define

$$F_k(\mathcal{T}_T) = F^{(\mu, h_k)}(\mathcal{T}_T)$$
$$= -\int_0^T \lambda_k(s) \, \mathrm{d}s + \sum_{T_i \in \mathcal{T}_T} \log(\lambda_k(T_i)).$$

We also consider

$$\pi_{\mathbf{p}, \mu, \mathbf{h}}^k(.) := \phi_k^{\mathbf{p}}(F^{\mu, \mathbf{h}}(.)), \tag{3}$$

with the $\phi_k^{\mathbf{p}}$'s defined in the same manner of the $\phi_k^{\mathbf{p}^*}$'s given in Proposition 2.1. Finally, we denote $\boldsymbol{\pi}_{\mathbf{p}, \mu, \mathbf{h}}(.) = \left(\pi_{\mathbf{p}, \mu, \mathbf{h}}^k(.)\right)_{k \in \mathcal{Y}}$ and $\pi := \boldsymbol{\pi}_{\mathbf{p}, \mu, \mathbf{h}}$.

A plug-in type classifier $g_\pi$ is naturally defined as

$$g_\pi(\mathcal{T}_T) = \underset{k \in \mathcal{Y}}{\operatorname{argmax}} \ \pi^k(\mathcal{T}_T). \tag{4}$$

## 3.3 PROPERTIES

In this section, we establish important properties of plug-in type classifiers. For a vector of functions $\mathbf{h} \in \mathcal{H}^K$, let us denote the supremum norm

$$\|\mathbf{h}\|_{\infty, T} = \max_{k \in \mathcal{Y}} \sup_{t \in [0, T]} |h_k(t)|.$$

We introduce for a positive constant $A$ the following set

$$\mathcal{H}_A^K := \left\{ \mathbf{h} \in \mathcal{H}^K \text{ s.t. } \sup_{\mathbf{h} \in \mathcal{H}^K} \|\mathbf{h}\|_{\infty, T} \leq A \right\}$$

and the set of probabilities

$$\Pi = \left\{ \boldsymbol{\pi}_{\mathbf{p}, \mu, \mathbf{h}} : \ \mathbf{p} \in \mathcal{P}_{p_0}, \ \mu \in (\mu_0, \mu_1), \ \mathbf{h} \in \mathcal{H}_A^K \right\}. \tag{5}$$

The first result is a key step to obtain the consistency of the classification procedure presented in Section 4.

**Proposition 3.4.** *Let us consider $\pi$ and $\pi^{'}$ two vectors functions belonging to the set $\Pi$ defined by (5) with respective parameters $(\mathbf{p}, \mu, \mathbf{h})$, and $(\mathbf{p}^{'}, \mu^{'}, \mathbf{h}^{'})$. Grant Assumptions 3.1, 3.2, 3.3, the following holds*

$$\mathbb{E}\left[\left\|\pi - \pi^{'}\right\|_1\right] \ \leq \ C\left(\left|\mu - \mu^{'}\right| + \left\|\mathbf{h} - \mathbf{h}^{'}\right\|_{\infty, T} \right.$$
$$\left. + \left\|\mathbf{h} - \mathbf{h}^{'}\right\|_{\infty, T}^2 + \left\|\mathbf{p} - \mathbf{p}^{'}\right\|_1\right),$$

*where $C$ is a constant depending on $K$, $T$, $\mathbf{h}^*$, $\mu_0$, $\mu_1$, $p_0$ and $A$.*

Proposition 3.4 provides a bound on $L_1$-distance between two elements of the set $\Pi$. It shows that this distance is bounded by the distance between the corresponding parameters of the associated models. From this result, for a plug-in type classifier $g$, we can easily deduce a bound of its excess risk.

**Corollary 3.5.** *For all $\pi = \pi_{\mathbf{p},\mu,\mathbf{h}} \in \Pi$, we have that*

$$
\begin{aligned}
\mathcal{E}(g_\pi) \;\leq\; & C\Big(|\mu - \mu^*| + \|\mathbf{h} - \mathbf{h}^*\|_{\infty,T} \\
& + \|\mathbf{h} - \mathbf{h}^*\|_{\infty,T}^2 + \|\mathbf{p} - \mathbf{p}^*\|_1\Big),
\end{aligned}
$$

*where $C$ is a constant depending on $K$, $T$, $\mathbf{h}^*$, $\mu_0$, $\mu_1$, $p_0$ and $A$.*

An important consequence of this result is that a plug-in type classifier which relies on consistent estimators of $\mathbf{p}^*$, $\mu^*$ and $\mathbf{h}^*$ is then consistent *w.r.t.* misclassification risk.

# 4 CLASSIFICATION PROCEDURE

This section is devoted to the presentation and the study of the proposed data-driven procedure that mimics the Bayes classifier. Our estimation method is then presented in Section 4.1 and theoretical guarantees of the procedure are derived in Section 4.2.

## 4.1 ESTIMATION STRATEGY

Based on the results of Section 3, we propose an hybrid classification procedure which involves both plug-in and empirical risk minimization (E.R.M.) principles. To this end, we introduce a learning sample $\mathcal{D}_n = \{(\mathcal{T}_T^i, Y^i), i = 1, \dots, n\}$, which consists of $n$ independent copies of $(\mathcal{T}_T, Y)$.

We propose a two-steps procedure. In a first step, we estimate the vector $\mathbf{p}^*$ by its empirical counterpart $\widehat{\mathbf{p}}$. The second step relies on the empirical risk minimization over a suitable set. In view of the results obtained in Section 3.3, we introduce the following approximation of the set $\Pi$:

$$
\widehat{\Pi} = \big\{\boldsymbol{\pi}_{\widehat{\mathbf{p}},\mu,\mathbf{h}} : \; \mathbf{p} \in \mathcal{P}_{p_0}, \; \mu \in (\mu_0,\mu_1), \; \mathbf{h} \in \mathcal{H}_A^K\big\} \quad (6)
$$

and the corresponding set of classifiers:

$$
\mathcal{G}_{\widehat{\Pi}} = \{g_\pi : \; \pi \in \widehat{\Pi}\}.
$$

Since $g^*$ is the minimizer of the misclassification risk, a natural estimator of $g^*$ would be the empirical risk minimizer over the family $\mathcal{G}_{\widehat{\Pi}}$

$$
\hat{g} = \underset{g \in \widehat{\Pi}}{\operatorname{argmin}} \; \frac{1}{n} \sum_{i=1}^{n} \mathbb{1}_{\{g(\mathcal{T}_T^i) \neq Y^i\}}.
$$

Nevertheless, as a solution of non convex minimization problem, it is known that this estimator is computationally intractable.

**Convexification**  To avoid computational issues, it is then natural to replace the classical 0-1 loss with a convex surrogate (see [Zhang, 2004]). Let us denote the scores functions set:

$$
\mathcal{F} := \{\mathbf{f} = (f^1, \dots, f^K) : \cdot \to \mathbb{R}^K\}.
$$

As convex surrogate, we consider the square loss and then define for a score function $\mathbf{f}$, the following risk measure

$$
\mathcal{R}(\mathbf{f}) := \mathbb{E}\left[\sum_{k=1}^{K} \left(Z_k - f^k(\mathcal{T}_T)\right)^2\right],
$$

with $Z_k = 2\mathbb{1}_{\{Y=k\}} - 1$.

The choice of the square loss as a convex surrogate is motivated by the fact that, if we define $g(\cdot) = \underset{k \in \mathcal{Y}}{\operatorname{argmax}} \, f^k(\cdot)$, then

$$
\mathbb{E}\left[\mathcal{R}(g) - \mathcal{R}(g^*)\right] \leq \frac{1}{\sqrt{2}}\big(\mathbb{E}\left[\mathcal{R}(\mathbf{f}) - \mathcal{R}(\mathbf{f}^*)\right]\big)^{1/2}, \quad (7)
$$

with $f^{*k}(\mathcal{T}_T) = 2\pi_k^*(\mathcal{T}_T) - 1$ which satisfies $\mathbf{f}^* \in \underset{\mathbf{f} \in \mathcal{F}}{\operatorname{argmin}} \, \mathcal{R}(\mathbf{f})$. Hence, consistent procedure *w.r.t.* to the $L_2$-risk involves consistent classification procedure *w.r.t.* the misclassification risk.

**Resulting estimator**  As suggested by the form of the optimal score function $\mathbf{f}^*$, we then consider the set of scores functions

$$
\widehat{\mathcal{F}} = \{2\pi - 1 : \; \pi \in \widehat{\Pi}\},
$$

and then consider the empirical risk minimizer over $\widehat{\mathcal{F}}$:

$$
\widehat{\mathbf{f}} \in \underset{\mathbf{f} \in \widehat{\mathcal{F}}}{\operatorname{argmin}} \; \widehat{\mathcal{R}}(\mathbf{f}), \quad (8)
$$

with

$$
\widehat{\mathcal{R}}(\mathbf{f}) := \frac{1}{n} \sum_{i=1}^{n} \sum_{k=1}^{K} \left(Z_k^i - \mathbf{f}(\mathcal{T}_T^i)\right)^2. \quad (9)
$$

Finally, the resulting classifier $\widehat{g}$ is the plug-in type classifier associated to $\widehat{\mathbf{f}}$ defined as

$$
\widehat{g} = \underset{k \in \mathcal{Y}}{\operatorname{argmax}} \; \widehat{\mathbf{f}}^k. \quad (10)
$$

Note that, in order to reduce the computational burden, we have chosen to not introduce the estimation of the probability weights $\mathbf{p}^*$ in the minimization problem given in Equation (8). Nevertheless it remains a possible strategy.

In the next section, we establish rates of convergence of our classification procedure.

## 4.2 RATES OF CONVERGENCE

The study of the statistical performance of $\widehat{g}$ defined by (10) relies on the following assumption.

**Assumption 4.1.** *Let $\varepsilon > 0$, we assume that there exists a $\varepsilon$-net $\mathcal{H}_\varepsilon \subset \mathcal{H}_A^K$, w.r.t. sup-norm $\|\cdot\|_{\infty,T}$ such that*

$$
\log(\mathcal{C}_\varepsilon) \leq C \log\left(\varepsilon^{-d}\right),
$$

*where $\mathcal{C}_\varepsilon$ is the number of elements of $\mathcal{H}_\varepsilon$, $d \geq 1$ and $C$ is a positive constant which does not depend on $\varepsilon$.*

**Theorem 4.2.** *Grant Assumptions 3.1, 3.2 and 3.3 and Assumption 4.1. If $\mathbf{h}^* \in \mathcal{H}_A^K$, the following holds*

$$\mathbb{E}\left[\mathcal{R}(\widehat{g}) - \mathcal{R}(g^*)\right] \leq C \left(\frac{d \log(n)}{n}\right)^{1/4},$$

*where $C > 0$ depends on $K$, $T$, $\mathbf{h}^*$, $\mu_0$, $\mu_1$, $p_0$ and $A$.*

Theorem 4.2 establishes that, when $n$ goes to infinity, the proposed classification procedure is consistent provided that $\mathbf{h}^*$ belongs to $\mathcal{H}_A^K$. If $\mathbf{h}^*$ does not belong to $\mathcal{H}_A^K$, a classical additional bias term appears.

We also have to note that Theorem 4.2 applies for a broad class of functions $\mathcal{H}$. In particular, Assumption 4.1 covers the case where $\mathcal{H}$ is a bounded linear subspace of functions. Let $(\psi_j)_{j \geq 1}$ an orthonormal basis such that the basis functions are uniformly bounded and then we consider for $\theta_0 > 0$

$$\mathcal{H} = \left\{t \mapsto \left(\sum_{j=1}^d \theta_j \psi_j(t)\right)_+ : \quad \|\theta\|_2 \leq \theta_0\right\},$$

as Laguerre basis for example. Another important example is the parametric exponential family

$$\mathcal{H} = \{t \mapsto \alpha\beta \exp(-\beta t), \ 0 < \alpha < 1, \ 0 < \beta \leq \beta_0\},$$

with $\beta_0 > 0$. Finally, it is possible to obtain better rate of convergence when the estimation of the probability weights and the estimation of $(\mu^*, \mathbf{h}^*)$ are performed on two different independent datasets, this is the purpose of the next paragraph.

**Alternative strategy** Hereafter, we consider an alternative strategy. First, we split the dataset $\mathcal{D}_n$ into two independent samples $\mathcal{D}_n^1$ and $\mathcal{D}_n^2$. Fore sake of simplicity, we assume that $n$ is even and that the two datasets $\mathcal{D}_n^1$ and $\mathcal{D}_n^2$ have same size $n/2$. Based on $\mathcal{D}_n^1$, we estimate $\mathbf{p}^*$, and based on $\mathcal{D}_n^2$ we estimate $\mathbf{f}^*$. The resulting classifier $\widehat{g}$ satisfies the following theorem.

**Theorem 4.3.** *Grant Assumptions 3.1, 3.2, 3.3 and 4.1. If $\mathbf{h}^* \in \mathcal{H}_A^K$, we have*

$$\mathbb{E}\left[\mathcal{R}(\widehat{g}) - \mathcal{R}(g^*)\right] \leq C \left(\frac{d \log(n)}{n}\right)^{1/2},$$

*with $C > 0$ a numerical constant.*

Therefore, the classifier $\widehat{g}$ achieves parametric rate of convergence up to a logarithmic term. Note that from practical point of view, the splitting of the sample does not affect the performance of the classifier $\hat{g}$. Therefore, we do not consider this strategy in the numerical section.

## 4.3 COMMENTS

In this section we make comments about the proposed procedure.

**Parameter $\mu$** Contrary to the parameter $p_0$, the estimation procedure requires the knowledge of $\mu_0$ and $\mu_1$. This assumption is important to obtain the consistency property. However, we shall show in Section 5 that the procedure has good performance if we only assume that $\mu^* > 0$.

**Estimation of the weights $\mathbf{p}^*$** For the estimation of the mixture weights, another approach is to include the estimation of $\mathbf{p}^*$ in the minimization procedure. In this case, the rate of convergence of the classification procedure is the same as the one provided in Theorem 4.3. However, we do not consider this approach since it significantly increases the computational cost of the procedure, especially if the number of classes is large.

**Other approach** Another strategy is possible motivated by Proposition 3.4. For example, assuming that the triggering kernels belong to the exponential kernel family, then classical estimators of the parameters can be used. Therefore, with these estimators we can compute a plug-in type classifier. For this task, the methods implemented in the `tick` library as Maximum Likelihood or Least-Squares estimator can be used. In the next section we illustrate this strategy with the Least-Squares estimator.

## 5 NUMERICAL EXPERIMENTS

In this section, we present numerical experiments to illustrate the performance of the procedure described in Section 4.1 and refer to the resulting algorithm as `ERM`. We focus on the case where the set $\mathcal{H}$ is the parametric exponential family. Then our method is compared to the plug-in strategy presented in Section 4.3 which is referred as `PI`.

We also include a comparison with Long Short-Term Memory (`LSTM`) algorithm. Indeed, these recurrent neural networks are used in time series forecasting and are a natural solution to study time dependency in data. The main numerical limitation is that the user needs to choose a length for the data whereas in the case of point processes the length is different for each sequence. This length has consequently to be chosen large enough to not lose information on the test sample. We use the `tensorflow.keras` library of Python with tuning parameters calibrated as follows: `batch_size=10`, `epochs=100` and `learning_rate=0.01`.

The details of the implementation of the `ERM` estimator are given in Section 5.1. Then, we describe the experimental setting in Section 5.2 and discuss the obtained results in Section 5.3. The source code we used to perform the

experiments can be found at `https://github.com/charlottedion/HawkesClassification`.

## 5.1 IMPLEMENTATION

We present the implementation of our classification procedure in the case where the set of kernel functions $\mathcal{H}$ is the parametric exponential family defined as

$$\mathcal{H} = \{t \mapsto \alpha\beta \exp(-\beta t), \ 0 < \alpha < 1, \ \beta > 0\}.$$

We define for $\alpha, \beta \in \mathbb{R}$ the function

$$h_{\alpha,\beta}(t) = \text{expit}(\alpha)\exp(\beta)\exp(-\exp(\beta)t),$$

where expit denotes the inverse-logit function. Then, we can write $\mathcal{H}$ as $\mathcal{H} = \{t \mapsto h_{\alpha,\beta}(t), \ \alpha, \beta \in \mathbb{R}\}$. For $\boldsymbol{\alpha}$ and $\boldsymbol{\beta}$ in $\mathbb{R}^K$, we denote by $\mathbf{h}_{\boldsymbol{\alpha},\boldsymbol{\beta}}$ the corresponding function of $\mathcal{H}^K$. Therefore the set $\widehat{\Pi}$ defined in Equation (6) can be rewritten as

$$\widehat{\Pi} = \{\boldsymbol{\pi}_{\hat{\mathbf{p}},\exp(\mu),\mathbf{h}_{\boldsymbol{\alpha},\boldsymbol{\beta}}}, \ \mu \in \mathbb{R}, \ \boldsymbol{\alpha}, \boldsymbol{\beta} \in \mathbb{R}^{\mathbb{K}}\}.$$

Hence the minimization step is performed *w.r.t.* $\mu$, $\boldsymbol{\alpha}$, and $\boldsymbol{\beta}$. Note that the formulation of the above set $\hat{\Pi}$ shows that the optimization part of our classification procedure does not require any constraint on the parameters. The minimization is performed with the `Python` function `minimize` with argument method `BFGS`. Algorithm 1 sums up the main steps of the procedure.

---

**Algorithm 1** Classification algorithm

---

**Input:** $T$, $\mathcal{D}_n$, and new observation $\mathcal{T}_{n+1}$
    Estimate $\mathbf{p}^*$ on $\mathcal{D}_n$
    Solve the minimization problem (8) based on $\mathcal{D}_n$
    Compute $\widehat{g}$ the resulting classifier (10)
    Compute $\widehat{Y}_{n+1} = \widehat{g}(\mathcal{T}_{n+1})$
**Output:** Predicted label $\widehat{Y}_{n+1}$

---

For the procedure `PI`, we use the `tick` function `HawkesExpKern` with argument `gofit = least-squares` for the parameter inference.

## 5.2 EXPERIMENTAL SETTING

We consider $K = 2$ or $K = 3$ classes in the following. We propose two different models for the experiments that we refer to as Model 1 and Model 2. For Model 1, we consider the case where the triggering kernel belongs to the parametric exponential family. For Model 2, we investigate a more general form for the kernels (see below). We set the baseline intensity $\mu = 1$. We use the library `tick` to generate the sequence of jump times of the Hawkes processes.

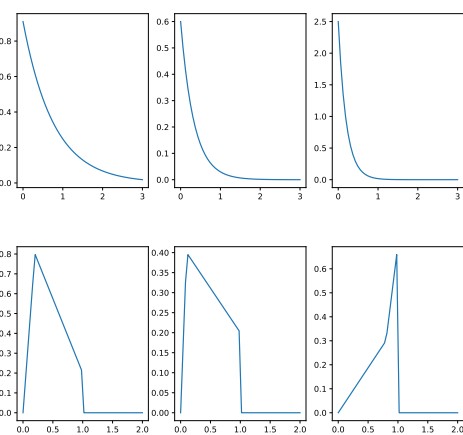

Figure 1: Kernel functions of Top: Model 1 and Bottom: Model 2 for Left: class $Y = 1$, Middle: class $Y = 2$ and Right: class $Y = 3$.

**Synthetic data** The label $Y$ is drawn from a uniform distribution on $\{1, \ldots, K\}$. Conditionally on $Y$, we simulate the jump times according to Model 1 and Model 2 which are defined as follows:

**Model 1** exponential kernels $h(t) = \alpha\beta \exp(-\beta t)$, with $(\alpha, \beta) = (0.7, 1.3)$ for class $Y = 1$, $(0.2, 3)$ for class $Y = 2$, and if $K = 3$, $(0.5, 5)$ for class $Y = 3$.

**Model 2** interpolation function kernels with parameters $(a, b, c)$:

$$h(t) = \begin{cases} \frac{b}{a}t, \ t \in [0, a], \\ \frac{b-c}{a-1}t + (b - \frac{b-c}{a-1}a), t \in ]a, 1[ \\ 0, \ t \geq 1 \end{cases}$$

with for $(a, b, c) = (0.2, 0.8, 0.2)$ for class $Y = 1$, $(0.1, 0.4, 0.2)$ for $Y = 2$, and if $K = 3$, $(0.8, 0.3, 0.7)$ for class $Y = 3$.

As an illustration, Figure 1 displays the considered kernels for both models. We can see from this figure that for Model 1 the kernel of the class $Y = 1$ seems to be different of the kernels of the classes $Y = 2$ and $Y = 3$ which are more closed. Hence, it should be easy to discriminate between observations from class $Y = 1$ and observations from class $Y \in \{2, 3\}$. On the contrary, observations from class $Y = 2$ and class $Y = 3$ would be overlapped. Similar comments can be made for Model 2 with observations from class $Y \in \{1, 2\}$ and observations from class $Y = 3$.

We also investigate the role of parameter $T$ on the difficulty of classification problem. To this end, Figure 2 displays the error rate of the Bayes classifier as a function of $T$ for Model 1 and $K = 3$. This error quickly decreases from 0.3 to 0.05 as $T$ goes from 10 to 40. In the following, we shall give results for $T = 20$.

Table 1: Error rates of Bayes, ERM, PI, and LSTM classifiers for $n = 100$, $T = 20$.

| CLASSIFIER: | BAYES | ERM | PI | LSTM |
|---|---|---|---|---|
| $K = 2$, MODEL 1 | 0.07 (0.01) | 0.08 (0.01) | 0.08 (0.01) | 0.09 (0.01) |
| $K = 2$, MODEL 2 | 0.27 (0.01) | 0.29 (0.02) | 0.29 (0.01) | 0.33 (0.02) |
| $K = 3$, MODEL 1 | 0.17 (0.01) | 0.18 (0.02) | 0.19 (0.02) | 0.36 (0.03) |
| $K = 3$, MODEL 2 | 0.39 (0.01) | 0.46 (0.02) | 0.45 (0.02) | 0.54 (0.03) |

Table 2: Error rates of Bayes, ERM, PI, and LSTM classifiers for $n = 1000$, $T = 20$.

| CLASSIFIER: | BAYES | ERM | PI | LSTM |
|---|---|---|---|---|
| $K = 2$, MODEL 1 | 0.07 (0.01) | 0.08 (0.01) | 0.08 (0.01) | 0.08 (0.01) |
| $K = 2$, MODEL 2 | 0.27 (0.01) | 0.28 (0.01) | 0.28 (0.02) | 0.30 (0.01) |
| $K = 3$, MODEL 1 | 0.17 (0.01) | 0.17 (0.01) | 0.18 (0.01) | 0.33 (0.02) |
| $K = 3$, MODEL 2 | 0.39 (0.01) | 0.43 (0.01) | 0.44 (0.01) | 0.49 (0.02) |

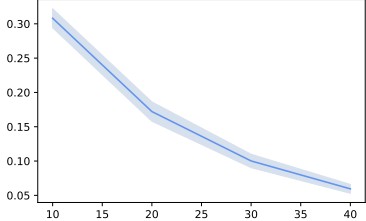

Figure 2: Error rate of the Bayes classifier as a function of $T$ for $K = 3$, $n = 100$.

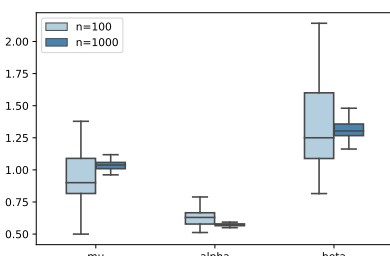

Figure 3: Boxplots of estimates of $(\mu, \alpha, \beta)$ of Model 1 for class $Y = 1$ for 50 repetitions. True parameters are $(1, 0.7, 1.3)$.

**Simulation scheme** In order to assess the performance of our procedure, we evaluate the misclassification risk of the Bayes classifier, ERM, PI, and LSTM through Monte-Carlo repetitions. More precisely, for $n \in \{100, 1000\}$ and $n_{\text{test}} = 1000$, we repeat independently 50 times the following steps:

1. simulate two datasets $\mathcal{D}_n$ and $\mathcal{D}_{n_{\text{test}}}$,

2. from $\mathcal{D}_n$ compute the classifier $\widehat{g}$, and

3. based on $\mathcal{D}_{n_{\text{test}}}$, compute the empirical error rate of the three classifiers.

The obtained results are presented in Table 1 for $n = 100$ and Table 2 for $n = 1000$. Note that, for ERM algorithm, the following initial guess for the optimization step is considered: $\mu = 0.5$, $\alpha = 1$ and $\beta = 1$ for all classes.

## 5.3 RESULTS

From the obtained results, we make several comments. For Model 1, the ERM and PI procedures achieve similar performance to the Bayes classifier for $n \in \{100, 1000\}$, and $K \in \{2, 3\}$. Similar comments can be made for Model 2

and $K = 2$. For $K = 3$, the difference between the error rates of the Bayes classifier and the ERM and PI classifiers is larger. Let us notice that, when $n$ increases the error rate of ERM is closer to the error rate of the Bayes classifier.

Finally, we can see that the LSTM algorithm has the worst performance in almost every scenario except for model 1 and $K = 2$. Hence, our classifier ERM is competitive to classify event sequences and can be recommended for future works.

Let us notice that our procedure also outputs estimations of the parameters $(\mu, \alpha, \beta)$. Although the estimation task is not our main purpose, it is interesting to evaluate the accuracy of the obtained estimators. Figure 3 displays a visual description of the obtained estimates for $n \in \{100, 1000\}$ for Model 1 with observations coming from the class $Y = 1$. Again, we can see the impact of the parameter $n$. For $n = 1000$, the estimation of the three parameters are clearly better than for $n = 100$. Furthermore, for $n = 1000$, the resulting estimates are quite good.

# 6 DISCUSSION

We investigate the multiclass classification setting where the features come from a mixture of simple linear Hawkes processes. In this framework, we derive the optimal predictor and provide a classification procedure tailored to this problem. The resulting algorithm relies on both plug-in and empirical risk minimization principles. We establish theoretical guarantees and illustrate the good performance of the method through a numerical study.

In future works, we plan to extend our classification procedure to the case where the observations come from a mixture of multidimensional Hawkes processes. Indeed, in neuroscience, the modeling of multivariate neuron spike data is used for taking into account potential interactions between neurons (see *e.g.* [Hansen et al., 2015], [Donnet et al., 2020]). Hence, it should capture the interactions between neurons. In this framework, a challenge is to take into account the high dimension of the space of parameters. For example, by considering exponential kernels, plug-in type classifier should benefit from algorithm as ADM4 which is adapted for high dimensional setting [Bacry et al., 2020].

Another possible development is the case of nonlinear Hawkes process. A few works focus on this subject, see *e.g.* [Brémaud and Massoulié, 1996], [Lemonnier and Vayatis, 2014], [Costa et al., 2020]. This allows us to consider kernels which can take negative values to model an inhibitory behaviour. The proposed algorithm should remains efficient. Nevertheless, it will be trickier to establish rates of convergence.

Finally, we could also extend our method to a model with a common time-inhomogeneous baseline. This idea is considered in many applications (see *e.g.* [Li et al., 2017]) and could be an improvement of the present algorithm.

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
