# OpenReview forum: "Multiclass Classification for Hawkes Processes"
_auai.org/UAI/2022/Conference — UAI 2022 Poster_

### Official Review · Reviewer_tpfN · 2022-04-08

**Q2(1) Originality/Novelty:** 2
**Q2(2) Significance/Impact:** 2
**Q2(3) Correctness/Technical Quality:** 3
**Q2(6) Clarity Of Writing:** 2
**Q6 Overall Score:** 6
**Q8 Confidence In Your Score:** 1

**Q1 Summary And Contributions:**

The paper proposes a method for multi-class classification problem in which the features are sequences of events. It is assumed that the data a generated by a mixture of linear Hawkes process and the paper derives an optimal Bayes rule with a two-step estimation procedure of the Bayes classifier. Theoretical guarantees for consistency are provided and rates of convergency are derived.

**Q10 Ethical Concerns (Optional):**

No.

**Q2 Assessment Of The Paper:**

More detailed information regarding each of these aspects is given below:

**Q2(4) Quality Of Experiments (Optional):**

2: Fair: The experimental evaluation is weak: important baselines are missing, or the results do not adequately support the main claims.

**Q2(5) Reproducibility:**

3: Good: Key resources (e.g., proofs, code, data) are available and key details (e.g., proofs, experimental setup) are sufficiently well-described for competent researchers to confidently reproduce the main results.

**Q3 Main Strengths:**

The paper has theoretical guarantees for consistency and derives rates of convergence (albeit, I found it hard to check the correctness of these proofs).

The method is interesting and relevant to the problem of classification with temporal dependence in the features. The idea overall appears to be novel.

**Q4 Main Weakness:**

At times I found the paper hard to follow. It is not a trivial method with extensive notation, extra effort in making the proofs/general ideas accessible could increase the impact of the paper.

The experimental results are rather limited: only simulated data sets are considered with maximum 3 classes. It would have been interesting if the paper also included PHEME data set: more classes where some classes are unbalanced. This also would place the paper in the literature as this is a common data set used in possibly competitive methods (Lukasik et al. 2016, Dutta et al. 2020, Tondulkar et al. 2020,...),.



**Q5 Detailed Comments To The Authors:**

Tables 1 and 2 states that these are classification accuracies. Aren't these errors, i.e. the smaller number in the table the better? I found the titles of the tables confusing.



**Q7 Justification For Your Score:**

I appreciate that the paper goes into the trouble to provide relevant theory and proofs, but unfortunately without more detailed comments I could not fully check the correctness. I do not fully put the blame on the presentation, it also could be because this is far from what I read on day to day basis. But I believe with more details it can be followed by a general CS reader. Additionally, the experiments seem to be rather weak and do not place the paper clearly in the literature.



**Q9 Complying With Reviewing Instructions:**

1: Yes.

---

### Official Review · Reviewer_2jZD · 2022-04-11

**Q2(1) Originality/Novelty:** 2
**Q2(2) Significance/Impact:** 2
**Q2(3) Correctness/Technical Quality:** 3
**Q2(6) Clarity Of Writing:** 4
**Q6 Overall Score:** 5
**Q8 Confidence In Your Score:** 1

**Q1 Summary And Contributions:**

This paper explores multiclass classification problems in cases where the features are temporal. They mainly deal with supervised classification for Hawkes processes.


**Q2 Assessment Of The Paper:**

More detailed information regarding each of these aspects is given below:

**Q2(4) Quality Of Experiments (Optional):**

1: Poor: The experimental evaluation is flawed or the results fail to adequately support the main claims.

**Q2(5) Reproducibility:**

3: Good: Key resources (e.g., proofs, code, data) are available and key details (e.g., proofs, experimental setup) are sufficiently well-described for competent researchers to confidently reproduce the main results.

**Q3 Main Strengths:**

The paper is well written. The idea is neat and might be useful for many problems in Neuroscience such as spikes in neurons from different populations.


**Q4 Main Weakness:**

The experiments are only applied to the synthetic data.

**Q5 Detailed Comments To The Authors:**

Would be great if we could see the method is applied to a Neuroscience problem, for example.
Please refer to Q3 and Q4.

**Q7 Justification For Your Score:**

In the experiments, only simulated data is used. I would like to see the model applied to real data.


**Q9 Complying With Reviewing Instructions:**

1: Yes.

---

### Official Review · Reviewer_Hn5N · 2022-04-12

**Q2(1) Originality/Novelty:** 2
**Q2(2) Significance/Impact:** 2
**Q2(3) Correctness/Technical Quality:** 3
**Q2(6) Clarity Of Writing:** 4
**Q6 Overall Score:** 6
**Q8 Confidence In Your Score:** 4

**Q1 Summary And Contributions:**

Paper proves statistical performance bound on accuracy of classification task when examples are trajectories drawn from different Hawkes processes (each trajectory from one of the set).

**Q2 Assessment Of The Paper:**

More detailed information regarding each of these aspects is given below:

**Q2(4) Quality Of Experiments (Optional):**

2: Fair: The experimental evaluation is weak: important baselines are missing, or the results do not adequately support the main claims.

**Q2(5) Reproducibility:**

4: Excellent: Key resources (e.g., proofs, code, data) are available and key details (e.g., proof sketches, experimental setup) are comprehensively described for competent researchers to confidently and easily reproduce the main results.

**Q3 Main Strengths:**

+ Trajectory classification a useful and (relatively) unstudied problem from this point-of-view.
+ Clearly defines terms used
+ Provides rate of convergence result of classification in this domain

**Q4 Main Weakness:**

- base rates must be the same across all classes
- Thm 4.2 (main theorem) does not provide dependence on features of the problem other than number of trajectories (proofs do not shed light on it either).
- consistency (which this theorem does go beyond) was previously established through identifiability of kernels
- results do not show advantage of ERM or "plug-in" estimate
- comparison to LSTM is irrelevant.

**Q5 Detailed Comments To The Authors:**

I think it would be helpful in the introduction to make even more explicit that the classification is per trajectory (and not per event)

(minor) I would call phi a "softmax" as it compounds a p and an x term.  It just converts a log-likelihood and prior into a posterior.

(minor) I'm uncertain why the squared error compares 2z-1 and 2pi-1 instead of just z and pi

(minor) Assumption 4.1 and Theorem 4.2 (which are right next to each other) both use C, but (I think) for different quantities.

(major) A real contribution of this paper would be to show how Thm 4.2's result depends on T, h and the mu bounds.  Often, I can get more samples (n) or more length of time (T).  I have an expectation about how C should depend on these, but I'd like to see it mapped out.  Showing how the difficulty of the learning problem scales with mu and h would be very helpful.  My skimming of the proofs didn't suggest it could be gleamed from there, but perhaps I missed it?

Along similar lines, the Laguerre basis is a fine idea, but how do the properties of the basis contribute to the bound?

(minor) "can be write" => "can be written"

I find the comparison to LSTM silly.  I'm guessing this might be due to the request of someone to compare to a "DNN model," but given that the data are drawn from a HP and completely synthetic, I'm not sure the value of an LSTM comparison (especially as there are continuous-time HP versions that could easily be adapted to classification).

Figure 2 should be plotted on a log-log so that it's n^{1/4} or n^{1/2} dependence can be made clear

That the consistency of such an estimate has already been established (through the consistency of estimation of kernel parameters) should be mentioned and cited.


**Q7 Justification For Your Score:**

The theorem is new to my knowledge.  Using a HP roughly in the manner described to do classification is not a novel idea, but the analysis is.  The experimental results are weak (see comments above) but not the main contribution.  The theorem is fine, but could have more details (dependence of "C" on properties of the problem other than the number of examples, for instance).

**Q9 Complying With Reviewing Instructions:**

1: Yes.

---

### Official Review · Reviewer_sMbR · 2022-04-15

**Q2(1) Originality/Novelty:** 3
**Q2(2) Significance/Impact:** 3
**Q2(3) Correctness/Technical Quality:** 3
**Q2(6) Clarity Of Writing:** 3
**Q6 Overall Score:** 5
**Q8 Confidence In Your Score:** 4

**Q1 Summary And Contributions:**

The authors propose an efficient classification method based on the bayes rule and provide both consistency of   the resulting estimators as well their convergence rates. Experimental results are shown on a couple of  synthetic datasets.



**Q2 Assessment Of The Paper:**

More detailed information regarding each of these aspects is given below:

**Q2(4) Quality Of Experiments (Optional):**

2: Fair: The experimental evaluation is weak: important baselines are missing, or the results do not adequately support the main claims.

**Q2(5) Reproducibility:**

3: Good: Key resources (e.g., proofs, code, data) are available and key details (e.g., proofs, experimental setup) are sufficiently well-described for competent researchers to confidently reproduce the main results.

**Q3 Main Strengths:**

The paper is well-laid out with the major contributions provided in the main paper and the detailed proofs       delegated to the appendix. Some concerns remain on the estimation procedure and the experiments are a bit limited.

 (A) Multiclass classification of Hawkes is an important problem in various e-commerce, financial, and healthcare applications. However, consistency of the estimators seems to be an open issue. This paper provides   both convergence rates scaling in O(\sqrt(dlogn/n)) as well as consistent classification estimators
 (B) Experimental resuls show the benefits of the proposed approach over standard LSTM approaches.

**Q4 Main Weakness:**


 (1) Unclear how the estimation of the weights p can be decoupled from the estimation of Hawkes parameters.
 Typically, there is an EM approach to alternate over the two?

 (2) The experiments use LSTM as baseline method for comparison which may not be suitable for the task.



**Q5 Detailed Comments To The Authors:**

 * It would be interesting to visualize the resulting mixtures generated from the mixture of hawkes with the overlay of ground truth and the estimation.

 * Add comparisons to other approaches such as Neural Hawkes Process (https://arxiv.org/abs/1612.09328) and their related extensions.

 * Some of the writing could be tightened such as on page one, we have a phrase like "one can cite .." and        section called comments with the first line saying "we make comments..".  The discussion section reads more      like conclusions and future work.



**Q7 Justification For Your Score:**

I think the paper is solid overall but I am bit unclear on the evaluations and the impact of the estimation of the mixture weights vs the parameters of the Hawkes. Will update my scores once I hear back from the authors.

**Q9 Complying With Reviewing Instructions:**

1: Yes.

---

### Decision · Program_Chairs · 2022-05-15

**Decision:**

Accept (Poster)

**Comment:**

Meta Review: Pros: technically sound contribution with theoretical guarantees
Cons: experiments only with simulated data although papers claims concrete motivation from brain research
I lean towards acceptance.